# Making Sense from Experience: How a Sustainable Multi-Sensory Event Spurs Word-of-Mouth Recommendation of a Destination Brand

**Mónica Gómez-Suárez** *[ID] **and María Jesús Yagüe** [ID]

Finance and Marketing Department, Universidad Autónoma de Madrid, Cantoblanco, 28760 Madrid, Spain; maria.yague@uam.es
* Correspondence: monica.gomez@uam.es; Tel.: +34-91-497-4348

**Abstract:** The last decade has seen an exponential growth in published articles related to the influence of marketing events on destinations. However, there is still a need for empirical research about the effect that organized events built upon sensorial components have on different variables related to participants' attitudes and behaviors, as such events have the ability to provide unique experiences and emotions. Therefore, this research focuses on the impact of a sustainable multi-sensory event marketing that promotes the interests of the organizing service company (a marina brand), alongside those of the host location, by associating the brand destination with this specific activity. By surveying attendees to a summer event aimed at enhancing visits to an area in Palma de Mallorca (Spain) and by adopting structural equation modeling estimation, the study shows that people's positive valuations of the event had an impact on their word-of-mouth recommendation of the brand. Thus, visitors' emotional experience was tied to their post-visit brand attitudes and brand equity. Based on the results, the study makes practical suggestions for branding in a sustainable destination, especially in relation to incorporating experiential elements in company-organized special events.

**Keywords:** tourism experiences; tourist behavior; event management; word-of-mouth; emotions; tourist destination; event marketing; engagement; tourism; brand

## 1. Introduction

In light of the concerns and opportunities invoked by the coronavirus crisis, sustainable behaviors have become an issue of high priority for both scholars and stakeholders [1,2]. In fact, the COVID-19 pandemic could be a potential catalyst for an essential transformation of the tourism industry [3]. Facing a massive crisis, governments, companies and territories need to understand some particular mechanisms that could serve to promote sustainable destinations. That way, an emphasis on sustainability may generate the growth needed to recover from the fallout. In this sense, the pandemic disaster could be a chance to reformulate new business plans that cover the three dimensions of sustainable development—environmental, economic and socio-cultural—in order to produce a suitable equilibrium.

In this sense, company-organized events might be a key element in fostering the recovery of brand destinations. This kind of events combines companies' strategic planning around attracting new visitors and caring for residents' needs. By designing mixed policies that respect the health measures imposed by authorities due to COVID-19, while also paying special attention to sustainability issues, local destination brands are helping their territories surmount the crisis. These actions require controlling the number of attendees, respecting the environment, promoting local businesses and outlets, enhancing community culture and social life, etc.

The present study focuses on experiential event marketing as one expression of this balance between new tourists' attraction and local inhabitants' necessities. Since the first

event-related articles appeared in the 1970s, events have become a well-established theme, featuring prominently in the development and marketing plans of most destinations [4]. Consequently, this research stream has experienced an exponential growth in the last decade. However, many studies have been theoretical in nature—see, for instance [5–8]. Thus, there is still a need for empirical research about the impact of events, particularly those based on sensorial or experiential components.

Experiential marketing is an important trend in the current age of the experiential economy [9,10]. Hence, experiential event marketing might be an effective tool for creating remarkable sustainable experiences, helping travelers and residents internalize the experience as an organic part of themselves and establish a special emotional connection with the brand. In fact, it appears that the environment and atmosphere are conducive to positive feelings and/or spiritual recovery for both tourists [11] and residents in a certain territory. Thus, more research is needed to conceptualize the visitors' sustainable experience and understand its relationship with the other variables.

Experiential event marketing could also help increase the probability of people recommending a destination brand [12]. Yet the academic research on the relationship between event marketing and brand recommendation or word-of-mouth (WOM) is scarce. More empirical evidence is needed to reveal the impact of post-visit personal antecedents that arise from one's experience with the brand destination [13]. Hence, there is a need to build analytical models to explain the interaction between these variables, as well as the role of intermediate constructs such as brand experience, brand attitude and brand equity. This is precisely the purpose of this study, which addresses the following objectives:

- To evaluate the emotional experience lived by the attendees of a marketing event organized by a touristic brand in an affluent destination.
- To measure the sequential indirect effects that event attitude and the emotional experience have on the WOM brand recommendation.
- To propose future research lines based on the results, that serve to promote sustainable destination brands in the face of the challenges provoked by the pandemic crisis.

Therefore, the present paper makes a threefold contribution. First, it reinforces the theoretical work on the nature of experience by exploring what makes event experiences transformative. Second, it explores the post-event evaluations of experiences and their effects on future intentions in a real situation rather than through a field experiment. Third, it proposes managerial aspects that can promote a resilient sustainable destination.

We conducted an empirical study in a destination in Mallorca (Spain) that attracts affluent visitors, both residents and tourists. Spain, like other Mediterranean countries such as France or Italy, is one of the few countries in the world in which a double situation occurs: tourism is a basic sector of the country's economy and, at the same time, the country is a leading destination of world tourism [14]. In this sense, tourism represents a bedrock industry.

The study is based on a survey carried out to measure the effect of a summer event organized by a marina brand (we cannot include the name of the company for confidentiality reasons). It was run in close collaboration with the marketing department of the company that manages the marina. We measured the effects of this special event, organized each summer by the company on the destination brand. Therefore, one advantage of this study is its use of actual field data, rather than information obtained under laboratory conditions. Moreover, although the data were gathered before the coronavirus pandemic, the results will serve to underline guidelines for bolstering recovery once the world can return to a focus on sustainable consumption and production practices, as recommended by the Sustainable Development Goal #12 of the United Nations' 2030 Agenda.

The remainder of this paper is organized as follows: The study begins with the theoretical background, the proposed hypotheses and the conceptual model. Next, we report the methodology used for the empirical analysis, along with the obtained results. The paper concludes by summarizing the study's main conclusions, its implications and some avenues for further research.

## 2. Conceptual Framework and Hypotheses

Figure 1 synthesizes the theoretical model that we have developed for the previously formulated hypotheses, that will be empirically tested in the following section.

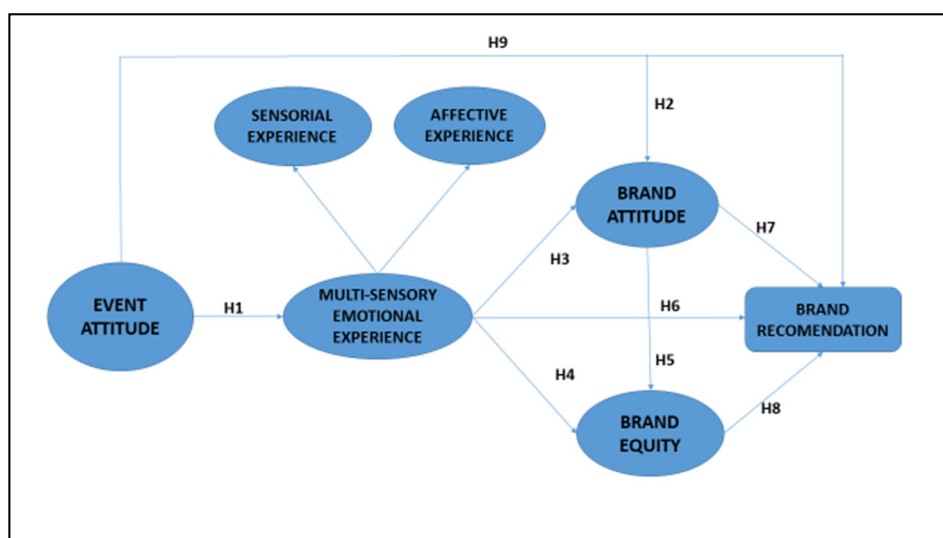

**Figure 1.** Theoretical Model.

Firstly, this conceptual framework is underpinned by the Stimulus-Organism-Response (S-O-R) model outlined by [15]. This model is based on the assumption that the sensory variables produce a holistic emotional experience that influences emotional responses to the environment itself [16]. In that sense, the S-O-R paradigm attempts to explain how atmospheric signals such as music, lighting and smell, among others, affect individuals' internal states and their external responses. In the case of current research, we focus on a multi-sensory event (the Stimulus—S) that motivates individuals' (the Organism—O) emotional reactions and behavioral intentions (the Response—R; in this case, recommending the event).

Secondly, we incorporate the Experience Economy Theory in light of a recent trend in marketing to create experiences or unique moments that engage the client in a personal way [17]. The experiential marketing concept was popularized by [18], who postulated that consumer experiences can be characterized by five strategic experience modules: sensory (sense), affective (feel), creative cognitive (think), physical/behaviors and lifestyles (act) and social-identity (social). His seminal work on the foundations of experiential marketing provided the basis for subsequent academic contributions, which were followed by a long program of research on experiential consumption and marketing. While these dimensions have undergirded the last 20 years of academic contributions in this domain, recent efforts have begun to identify the specific components or dimensions of experiential consumption that can be applied in services marketing [19,20].

The notion of experience creation and promotion has always been central to destination marketing practices [21]. Marketing strategies based on an experiential economy seem like a reasonable step, given that experience is a major factor differentiating leisure services from positive images and memories [22]. In short, experiential marketing seeks to create a personalized connection with the audience through direct and emotional involvement, thereby transforming the purchasing act into a credible and unforgettable experience [23]. This literature stream considers the consumers' experience to be the main focus of current marketing strategies [24,25].

However, the current literature has produced limited explanations regarding the components that mark an impressive visitor experience [26]. Moreover, while many past studies employed the term 'experience', they either did not operationally define it or employed vague or ambiguous statements in their research instrument, potentially

impacting the validity of their results. Furthermore, it is evident that consumer experiences are multi-dimensional. Nevertheless, the sensorial dimension has been particularly ignored (see Agapito et al. (2017) [27]), even though it may have important relationships with other dimensions of the experience.

Indeed, empirical studies underscore the pivotal role of sensory components [28] in terms of engaging and co-creating value with consumers [29]. Sensory perceptions are representative of the emotional experience's external factors, while the affective sensations that they generate are considered internal factors [30]. In fact, some papers have confirmed the relationship between the stimulation from the physical environment and consumers' emotional status [31]. With this theoretical assumption in mind, we can delineate two main sub-dimensions of what we call a multi-sensory emotional experience: the sensorial experience and the affective experience. The former represents external stimuli relating to the senses, while the latter reflects individuals' internal reactions.

The various components comprised in experiential events affect the degree to which they become an effective tool through the creation of consumer experiences that involve sensory, emotional, cognitive, behavioral and relational values, which substitute functional values [18]. Nonetheless, experiential marketing events create a space for the target audience to engage with the organization, its brands and its community [32]. By allowing for direct and highly interactive consumer-brand encounters, event marketing can produce outstanding brand experiences [33]. Hence, it is a valuable choice for building and strengthening relationships, especially for service brands that do not provide a tangible output to the consumer [34]. According to [35], experiential marketing events are live situations designed by the company for marketing purposes, primarily to communicate particular messages to target audiences. There are several types of events that companies can utilize, but experience events are currently the most popular way to achieve memorable relations with the target audience(s) [36]. In contrast with event sponsorship, where a company mainly pledges its support, event marketing entails that the company is responsible for the planning, organization, execution and control of the events [32].

Even if both communication strategies use events to achieve their respective goals, marketing events are self-staged while event sponsorship involves an independent third-party (i.e., an event organizer) staging the event and conveying the messages aimed at the target group [37]. The participants in the events are more receptive to the marketing messages and the images associated with the event than they would be to other marketing tools used by the company [38].

As a form of strategic communication preceding the consumer experience [39], events represent one way for brands to induce a positive effect on consumers' perceptions [40]. The event attempts to create a 'flow state experience' for attendees, which may involve surprise, novelty or challenge [35]. Thus, event marketing seems to play an important role in creating long-lasting brand experiences [41]. Events seem to have a greater impact on consumers' perception and sensations when they combine a good experience, a trained staff and the possibility of co-creating a 'show' [42,43].

Facing an increasingly global competition for visitor spending, places lean on events as fundamental marketing propositions [44] in order to help individuals feel an emotional connection toward a place or destination [12,45]. The creation and delivery of emotional experiences can create brand differentiation and influence sales, loyalty and brand promotion [46]. For destinations, event experiences become more important to attendees when there is a clear association between the event and the location [47]. On this basis, it is important to understand how individuals react when a brand interacts with them, as the experience could directly impact the process of consumption and produce value for both parties [48].

To the best of our knowledge, there is a lack of appropriate empirical models for assessing the relationship between event attitude and multi-sensory experience. Thus, our first step is to review studies related to conventional communication tools, such as advertising or promotions. For instance, advertising schemas are believed to aid in

sensory data processing efficiency and influence affective (i.e., attitudinal) and behavioral responses [49]. Therefore, if the attitude-toward-ad affects the effectiveness of advertising as a whole [50], then the attitude-toward-event should similarly shape the effectiveness of the event as a whole. The empirical literature also highlights the critical role of retailers' promotional activities in customer experience strategies [51].

The literature on event sponsorship can also be instructive in this regard. In this domain, event attitude has always been a fundamental construct for evaluating a sponsorship's effectiveness [45,52]. Ref. [53] postulated that attitudinal responses to a sponsorship message could directly influence consumers' reactions to the sponsor. Through personal interactions, experiential marketing allows consumers to connect emotionally with a sponsor and develop certain attitudes toward the type of event (e.g., sporting events or cultural events). For instance, [52] uncovered a relationship between positive (negative) emotions and positive (negative) event attitude in a large event such as the Olympic Games. Further, attendees associate emotions with certain activities comprised in the event experience [38]. In the same vein, one's attitude toward a particular event has a positive influence on the experience [23,40,54]. Considering these antecedents, we can formulate the following hypothesis:

**Hypothesis 1 (H1).** *Event attitude has a positive effect on the multi-sensory emotional experience.*

Brand attitude is defined as the extent to which a consumer has a favorable (or unfavorable) view of a brand [55]. This construct plays a pivotal role in the effectiveness of many different marketing and communications media [33]. Branding literature has largely focused on the relationships between event marketing and brand experience [33,34,56]. Past studies found a positive link between sponsorship and favorable brand perception [38,57,58]. In short, a more positive attitude toward the event generally aligns with a more positive brand attitude [37,59] and can also be associated with other characteristics, such as the product's category, nationality or degree of complementarity with other products. Each of these characteristics can help translate positive associations about the event to the brand sponsoring the event [60]. Because the literature has already established that marketing events can foster a positive attitude toward the involved brand, we propose the following empirical hypothesis:

**Hypothesis 2 (H2).** *Event attitude has a positive effect on brand attitude.*

Previous literature underlines the importance of emotional experiences in generating spontaneous positive attitudes toward a brand [61,62]. According to [63], the experiences are pieces of information that bring symbolic and experiential benefits that can favorably influence a person's attitude toward the associated brand(s) [64]. In the event literature, scholars have found some partial empirical results in terms of brand experience [56] and its relation to participants' attitude toward the event and the organizing/sponsoring brand(s) [33,41,59]. In the field of tourism, specifically, [65] uncovered a significant relationship between emotional experiences lived in mega-events and brand attitudes toward the city where they are held. From this evidence, we formulate the following hypothesis:

**Hypothesis 3 (H3).** *A multi-sensory emotional experience has a positive effect on brand attitude.*

Marketing events have the potential to create an extraordinary experience for the consumer. This experience is a useful way to build, change and reinforce a brand's image through its association with the qualities of the event [35]—or, in short, to foster brand equity. Managers do not directly use consumer-based brand equity scales; instead, they apply proxy measures related to brand equity's components [66]. However, events can create a deeper and more meaningful brand equity-building connection with consumers through the provided experiences [67,68]. According to [69], event marketing has been viewed as valuable in generating awareness to the brand and corporate images, but its

ability to communicate a more sophisticated, specific message or contribute to other aspects of brand equity has not been sufficiently studied. In the mass consumer goods context, the sensory and affective reactions cued by emotional experiences can generate a greater experiential value. These reactions are perceived as an antecedent of global brand equity [70] or of some brand equity component, such as image [71] or loyalty [72]. However, there are few empirical studies that investigate brand equity in the context of events. Those that exist are mainly related to the field of sponsorship, especially in the sports field, where researchers have analyzed the effect of experience on global brand equity [73–75] or one of its components, such as brand image [37,76–78]. As a final point on brand equity [33], stressed that the relationship between event marketing and brand equity has been largely under-researched, despite the importance and popularity attributed to such events. By focusing on brand equity, these authors aimed to understand the impact of events not only from a communication perspective, but also in terms of the overall brand strategy. Ultimately, they posited that experience is an antecedent of brand attitude, mediating the relationship between brand equity in all types of events. Building on those authors' work, we developed two hypotheses that derive from the inclusion of brand equity in the conceptual model:

**Hypothesis 4 (H4).** *A multi-sensory emotional experience has a positive effect on brand equity.*

**Hypothesis 5 (H5).** *Brand attitude has a positive effect on brand equity.*

One of the main goals of an experiential marketing event is to create buzz or consumer conversations about the brand; thus, we evaluated WOM as a potential outcome [35]. WOM represents the informal communications between consumers about the ownership, usage or characteristics of particular goods and services and/or their sellers [79]. While the academic literature has defined WOM in various ways [80], we have adopted the most restrictive approach by understanding WOM as a recommendation between people who already know one another (e.g., family, friends) [81], whereas online or electronic WOM (eWOM) often occurs among people who do not know one another [82]. This study focuses solely on the first construct.

Research on consumers' purchase process and brand choice has highlighted the importance of WOM over other sources of communication [83]: namely because it is perceived as credible and custom-tailored by people with no apparent self-interest in pushing a product [84]. As a first-degree information about a brand, WOM exerts considerable influence over consumers' intention to buy [85], which is largely why many organizations want to leverage WOM recommendations as a marketing tool [80].

Environments that enable experiences are more likely to create a strong WOM, as consumers gather more memorable and distinguished impressions to share [84,86]. When these experiences are emotionally impactful, consumers are more likely to spread WOM [79,87]. In fact, when consumers seek feedback from other customers about hedonic consumer products, they expect to find affective and sensory experiences [88], which serve to intensify the WOM [89]. WOM recommendations appear to be one of the main methods of measuring the effectiveness of experiential marketing at a theoretical level [7,69]. Nevertheless, to the best of our knowledge, this assumption has only received scarce attention from the hospitality literature, such as the empirical developments from [90–92]. According to [13], more empirical evidence is still needed to reveal the impact of post-visit personal antecedents, mainly derived from visitors' experience with the brand destination on outcomes, such as WOM. Thus, we propose that the brand experience will impose a direct effect on the WOM recommendation. Formally expressed:

**Hypothesis 6 (H6).** *A multi-sensory brand experience has a positive direct effect on brand recommendations (WOM).*

At the same time, the previous literature suggests that the effect of the brand experience on the WOM recommendation occurs indirectly, through one's attitude toward the brand as well as the brand equity. Indeed, there is sufficient evidence to propose that brand attitude [93–95] and brand equity [96,97] have a positive effect on brand WOM recommendations. Hence, we hypothesize:

**Hypothesis 7 (H7).** *Brand attitude has a positive effect on brand recommendations (WOM).*

**Hypothesis 8 (H8).** *Brand equity has a positive effect on brand recommendations (WOM).*

Finally, we are not aware of any empirical work that proves the relationship between event attitude and brand recommendation, with the exception of [90], which found that the event (a festival brand) and consumers' participation in social networks act as a WOM antecedent. Therefore, we hypothesize the following:

**Hypothesis 9 (H9).** *Event attitude has a positive effect on brand recommendations (WOM).*

### 3. Methods

Our goal was to empirically analyze the impact of a real event organized by a service brand. We specifically considered the impact of an experiential marketing event on WOM recommendations for the organizing brand, as well as the role of experience, attitude and equity arising from their brand event.

Our collaborative partner in this study was a private marina located in Mallorca that has about 500 moorings, many of them intended to accommodate yachts of great length. This company manages a wide range of lively annual events. Since 2013, this company has organized a summer event based on a program of varied shows that are designed to encourage overnight stays in the port. The event pays special attention to its environment, helping local businesses promote the place where they are located.

The event has evolved over the years, incorporating more experiential elements (music, lighting, color, decoration, etc.) in order to create a remarkable experience for both residents and tourists. The particular event object of this research took place in the port during the summer, from Wednesday to Sunday, 8:30 pm to 11:45 pm each day. The entire campus was free and opened to the public, and the participation was voluntary. The event featured a program of varied shows designed to liven up the summer nights of the port. It had colorful light bulbs and banners in the wind, as well as an appealing combination of styles, scenic languages and extravagant characters. The enveloping atmosphere of this unique marketplace offered an exceptional experience to the five senses. The design of the event resembled that of a spectacle, with a ring in the center and performances happening in the encircling area. The ticket office, which monitored the venue's capacity, utilized a vintage aesthetic. The visitors had two food and drink kiosks, as well as an area of chairs and tables where they could eat or rest. Several channels within the geographical scope of Mallorca were used to advertise this event: radio, brochure mailings, social network promotions, street marketing actions, signage in the port itself and vinyl signboards in the corridors of Palma de Mallorca airport.

Before the survey, we asked three tourism experts to assess the validity of the questionnaire. We then conducted a pilot study during the first days of July, using a sample of visitors from different nations who were assisting with the event. We checked that the questions were clearly formulated and that the scales were reliable (Appendix A). We measured event attitude with four items on a semantic-differential scale developed by [98]. The original scale ranged from −3 to 3, but we rescaled it to 1 to 7 to match the other questions. All the items related to multi-sensorial experience, brand equity and brand attitude were anchored on a seven-point Likert scale, ranging from 1 ('strongly disagree') to 7 ('strongly agree'). The multi-sensory emotional experience was based on the scale by [72], with three items for sensorial experience and three items for affective experience. The brand equity scale featured three items adapted from [99]. The brand attitude scale was de-

veloped by [100]. All three scales were additionally validated by [33]. Finally, we measured WOM recommendation with a single item that assessed the participants' probability of recommending the brand to a friend, colleague or relative. The measure was based on the scale originally developed by [101] and ranged from 0 ('not at all likely') to 10 ('extremely likely'). In this pilot study, the Cronbach's alpha of each construct exceeded 0.80.

The main fieldwork was handled by an external company located in Palma de Mallorca, which possesses its own Computer Assisted Telephone Interviewing (CATI) center and a network of Spanish-, English- and German-speaking interviewers, from 31 July to 21 September 2017. The surveyors were required to contact participants at least one week after they attended the event. The response rate was almost 20% (1344 contacts, 260 questionnaires completed); of these, 87.6% were Spanish, 58.9% were female and the average age was 42.3 (standard deviation: 10.86). It is important to highlight that this event was not organized solely for tourists, as the company is also interested in promoting local businesses and the port image to local residents. This is reflected in the fact that 83.5% of the participants were residents of Mallorca. Only 18.6% were first-time visitors to the port, while 58.6% of participants had known the place for more than three years.

## 4. Results

In terms of the descriptive results (Table 1), the event fosters a very favorable visitor attitude, since the items garnered a very high average: between 6.363 and 6.526 on a scale from 1 to 7, with low dispersion (less than 1) among the attendees. This reflects on the lowest variation coefficients (VC) for this construct among all the others. All of the remaining scores reflect that the experiential event is capable of generating outstanding scores for the brand—in this specific case, substantiated by sensory components rather than affective components. The average for the three sensorial experience items was higher (Mean = 5.226; Standard Deviation (SD) = 1.175, on a scale from 1 to 7) than the average for the three items related to affective experience (Mean = 4.865; DT = 1.494). However, VC in the case of affective experience indicates that dispersion reached the highest value, accounting for more than 0.3 in two of the variables. The brand achieved a favorable attitude (Mean = 5.887; ST = 1.132). Relating to brand equity, the three items exceeded the medium point (4), but the average for the three items presented a high dispersion (Mean = 4.929; SD = 1.506). This means that there is a high variability on the responses to these items, accounting for more than 0.3 in two items and 0.29 in the other one. The descriptive measures for the last item, related to WOM, show that people highly recommended the marina brand (Mean = 8.725, on a scale from 0 to 10; SD = 1.069).

The event might be better remembered and obtain better evaluations by those attendees who visited it on more than one occasion than by those who did so on one occasion. Since there were no significant differences in the assessment of the event between those who went to the event once and those who experienced it more times, the level of exposure was not sufficiently decisive for the subsequent assessment of the event.

Next, we examined common method variance (CMV) by using Harman's single-factor test [102]. A Principal Component Analyses (PCA) with varimax rotation for the 18 initial items revealed a four-factor structure (Table 2), indicating the absence of a general factor that could explain the variance. One item for sensorial experience (senso1: I find brand X interesting for discovering new experiences) was excluded in the following analyses for three reasons: it showed low communality, contributed less to the scale reliability (Cronbach's alpha was 0.847 when it was included and 0.876 when it was not included) and contributed more to the brand attitude factor than to the multi-sensory factor. The loadings that appeared in the final configuration matrix (17 items) with four factors accounted for 81.31% of the variance extracted with the PCA model.

<div align="center">

**Table 1.** Descriptive Statistics.

</div>

| Constructs | Description | Mean | SD | VC |
|---|---|---|---|---|
| Event Attitude | Unexciting-Exciting (event1) | 6.363 | 0.832 | 0.131 |
| | Boring-Stimulating (event2) | 6.422 | 0.871 | 0.146 |
| | Monotonous-Sensational (event3) | 6.403 | 0.893 | 0.149 |
| | Unappealing-Appealing (event4) | 6.526 | 0.774 | 0.129 |
| | Mean Event Attitude Items | 6.434 | 0.843 | 0.131 |
| Sensorial Experience | I find brand X interesting for discovering new experiences (senso1) | 5.510 | 1.162 | 0.243 |
| | Brand X makes a strong impression on my visual sense or other senses (senso2) | 5.163 | 1.124 | 0.264 |
| | Brand X appeals to my senses (senso3) | 5.015 | 1.175 | 0.277 |
| | Mean Sensorial Experience Items | 5.226 | 1.134 | 0.256 |
| Affective Experience | Brand X induces my feelings and sentiments (affec1) | 4.859 | 1.282 | 0.302 |
| | I have strong feelings for brand X(affec2) | 4.726 | 1.416 | 0.343 |
| | X is a brand which generates positive feelings to me (affec3) | 5.030 | 1.233 | 0.289 |
| | Mean Affective Experience Items | 4.865 | 1.494 | 0.308 |
| Brand Attitude | Brand X is good (att1) | 5.861 | 1.115 | 0.209 |
| | Brand X is pleasant (att2) | 6.006 | 0.979 | 0.170 |
| | Brand X is attractive (att3) | 5.816 | 1.16 | 0.219 |
| | Mean Brand Attitude Items | 5.887 | 1.132 | 0.193 |
| Brand Equity | Even if any other port in the island has the same features as X, I would prefer to go to brand X (be1) | 5.000 | 1.423 | 0.310 |
| | If there is another port in the island as good as brand X, I prefer to go to brand X (be2) | 4.927 | 1.320 | 0.294 |
| | If another port in the island is not different from brand X in any way, it seems smarter to go to brand X (be3) | 4.871 | 1.392 | 0.312 |
| | Mean Brand Equity Items | 4.929 | 1.506 | 0.306 |
| WOM recommendation | Probability to recommend brand X to a friend, colleague or relative (WOM_1) | *8.725* | 1.069 | 0.183 |

Note: For event attitude, the scale ranged from −3 to 3 and was rescaled to 1 to 7. For brand experience, attitude and equity, the scale ranges from 1 to 7. For WOM, the scale ranges from 0 to 10.

Afterwards, we sought to determine whether the measurement model adequately fit the data. We ran a first-order confirmatory factor analysis (CFA) with maximum likelihood estimation in Amos Graphics 26.0 for the measurement model, which adequately fit the data. Past theory and studies suggest that the dimensions are reflective constructs, meaning that the items are manifestations of the construct [103]. Table 3 shows the results of the first-order measurement model. The CFA results indicated an acceptable data fit: $\chi^2 = 169.79$; $\chi^2$/degrees of freedom (df) = 2.17; Root Mean Square Error of Approximation (RMSEA) = 0.06; Comparative Fit Index (CFI) = 0.970; Tucker Lewis Index (TLI) = 0.969. All indicators had statistically significant loadings (Li) ($p \leq 0.05$) greater than 0.60. Cronbach's alphas for all constructs were in the range of 0.876 to 0.928 and thus exceeded the critical value of 0.70 [104]. The composite reliability (CR) exceeded the critical value of 0.70, with values ranging from 0.869 to 0.933. The Average Variance Extracted (AVE) values were greater than the cut-off score (0.50).

**Table 2.** Principal Component Analyses (PCA) Results.

| Rotated Components Matrix | | | | |
| --- | --- | --- | --- | --- |
| | Component | | | |
| | **Multi-Sensory Experience** | **Event Attitude** | **Brand Attitude** | **Brand Equity** |
| Brand X induces my feelings and sentiments | 0.863 | | | |
| I have strong feelings for brand X | 0.804 | | | |
| Brand X is a brand which generates positive feelings to me | 0.770 | | | |
| Brand X appeal to my senses | 0.762 | | | |
| Brand X makes a strong impression on my visual sense or other senses | 0.681 | | | |
| Unexciting- Exciting | | 0.912 | | |
| Boring-Stimulating | | 0.908 | | |
| Monotonous-Sensational | | 0.906 | | |
| Unappealing-Appealing | | 0.892 | | |
| Brand X is good | | | 0.859 | |
| Brand X is attractive | | | 0.858 | |
| Brand X is pleasant | | | 0.796 | |
| If there is another port in the island as good as brand X, I prefer to go to brand X. | | | | 0.894 |
| Even if any other port in the island has the same features as brand X, I would prefer to go to brand X. | | | | 0.852 |
| If another port in the island is not different from brand X in any way, it seems smarter to go to brand X. | | | | 0.764 |
| Variance Extracted (%) | 24.61 | 22.44 | 17.60 | 16.65 |

**Table 3.** Reliability and Convergent Validity First-Order Model.

| Item | Construct | Li | R2 | 1-R2 | Alfa | CR | AVE |
| --- | --- | --- | --- | --- | --- | --- | --- |
| affec1 | Affective Experience | 0.926 | 0.857 | 0.143 | | | |
| affec2 | Affective Experience | 0.859 | 0.738 | 0.262 | 0.902 | 0.903 | 0.757 |
| affec3 | Affective Experience | 0.822 | 0.676 | 0.324 | | | |
| senso2 | Sensorial Experience | 0.828 | 0.686 | 0.314 | 0.876 | 0.873 | 0.775 |
| senso3 | Sensorial Experience | 0.930 | 0.865 | 0.135 | | | |
| att1 | Brand Attitude | 0.904 | 0.817 | 0.183 | | | |
| att2 | Brand Attitude | 0.877 | 0.769 | 0.231 | 0.928 | 0.875 | 0.702 |
| att3 | Brand Attitude | 0.721 | 0.520 | 0.480 | | | |
| Be2 | Brand Equity | 0.955 | 0.912 | 0.088 | | | |
| Be3 | Brand Equity | 0.885 | 0.783 | 0.217 | 0.907 | 0.928 | 0.812 |
| Be4 | Brand Equity | 0.860 | 0.740 | 0.260 | | | |
| Event1 | Event Attitude | 0.913 | 0.834 | 0.166 | | | |
| Event2 | Event Attitude | 0.907 | 0.823 | 0.177 | 0.936 | 0.937 | 0.788 |
| Event3 | Event Attitude | 0.890 | 0.792 | 0.208 | | | |
| Event4 | Event Attitude | 0.840 | 0.706 | 0.294 | | | |

Additionally, Table 4 displays the discriminant validity of all constructs [105]. The AVE of all constructs (principal diagonal) was higher than the square inter-construct correlation in all the cases.

**Table 4.** Discriminant Validity for First-Order CFA Model.

| | Affective | Sensorial | Brand Attitude | Brand Equity | Event Attitude |
|---|---|---|---|---|---|
| **Affective** | 0.757 | | | | |
| **Sensorial** | 0.621 | 0.775 | | | |
| **Brand Attitude** | 0.282 | 0.295 | 0.702 | | |
| **Brand Equity** | 0.257 | 0.252 | 0.305 | 0.812 | |
| **Event Attitude** | 0.075 | 0.053 | 0.051 | 0.015 | 0.788 |

Note: The convergent validities are shown on the diagonal, and the square of the correlations appears below the diagonal.

We then estimated a second-order model, which reflected on sensory experience and affective experience, in order to include the construct for multi-sensory emotional experience (Table 5). Goodness-of-fit indexes were satisfactory $\chi2 = 170.79$; $\chi2/df = 2.13$; RMSEA = 0.039; CFI = 0.995; TLI = 0.991). We assessed reliability, as well as convergent and discriminant validity, in a way similar to that of the first-order model (Table 5).

**Table 5.** Reliability and Convergent Validity Second-Order Model.

| Item Construct | Construct | Li | R2 | 1-R2 | CR | AVE |
|---|---|---|---|---|---|---|
| Affective | Multi-sensory emotional Experience | 0.893 | 0.797 | 0.203 | 0.882 | 0.789 |
| Sensorial | Multi-sensory emotional Experience | 0.883 | 0.780 | 0.220 | | |
| affec1 | Affective Experience | 0.944 | 0.891 | 0.109 | | |
| affec2 | Affective Experience | 0.846 | 0.716 | 0.284 | 0.902 | 0.754 |
| affec3 | Affective Experience | 0.810 | 0.656 | 0.344 | | |
| senso2 | Sensorial Experience | 0.815 | 0.664 | 0.336 | 0.881 | 0.789 |
| senso3 | Sensorial Experience | 0.956 | 0.914 | 0.086 | | |
| att1 | Brand Attitude | 0.908 | 0.824 | 0.176 | | |
| att2 | Brand Attitude | 0.938 | 0.880 | 0.120 | 0.933 | 0.824 |
| att3 | Brand Attitude | 0.876 | 0.767 | 0.233 | | |
| Be2 | Brand Equity | 0.957 | 0.916 | 0.084 | | |
| Be3 | Brand Equity | 0.884 | 0.781 | 0.219 | 0.928 | 0.812 |
| Be4 | Brand Equity | 0.859 | 0.738 | 0.262 | | |
| Event1 | Event Attitude | 0.913 | 0.834 | 0.166 | | |
| Event2 | Event Attitude | 0.906 | 0.821 | 0.179 | 0.937 | 0.788 |
| Event3 | Event Attitude | 0.890 | 0.792 | 0.208 | | |
| Event4 | Event Attitude | 0.841 | 0.707 | 0.293 | | |

The model achieved discriminant validity, since the AVE values were higher than the inter-construct correlations (Table 6).

Table 7 illustrates the structural equations model (SEM) results, which presented adequate goodness-of-fit indexes ($\chi2 = 65.158$; $p = 0.05$ $\chi2/df = 1.357$; CFI = 0.993; GFI = 0.964; AGFI = 0.933; TLI = 0.999; RMR = 0.055; RMSEA = 0.037). All the proposed relationships are positive and statistically significant, except for one (multi-sensory-WOM recommendation). Thus, we confirmed all the proposed hypotheses except for H6.

**Table 6.** Discriminant Validity for Second-Order Model.

|  | Multi-Sensory | Brand Attitude | Brand Equity | Event Attitude |
|---|---|---|---|---|
| **Multi-sensory** | 0.789 | | | |
| **Brand Attitude** | 0.366 | 0.824 | | |
| **Brand Equity** | 0.324 | 0.305 | 0.812 | |
| **Event Attitude** | 0.081 | 0.051 | 0.015 | 0.788 |

Note: The convergent validities are shown on the diagonal, and the square of the correlations appears below the diagonal.

**Table 7.** SEM Model Results.

| Construct | | Construct | Estimate |
|---|---|---|---|
| Multi-sensory | <— | Event Attitude | 0.299 *** |
| Brand Attitude | <— | Multi-sensory | 0.508 *** |
| Brand Attitude | <— | Event Attitude | 0.164 *** |
| Affective | <— | Multi-sensory | 0.847 *** |
| Sensorial | <— | Multi-sensory | 0.861 *** |
| Brand Equity | <— | Brand Attitude | 0.275 *** |
| Brand Equity | <— | Multi-sensory | 0.437 *** |
| WOM recommendation | <— | Brand Equity | 0.145 * |
| WOM recommendation | <— | Event Attitude | 0.387 *** |
| WOM recommendation | <— | Brand Attitude | 0.104 * |
| WOM recommendation | <— | Multi-sensory | 0.109 |

Note: *** significant at 99%; * significant at 5%.

Figure 2 graphically presents the final results.

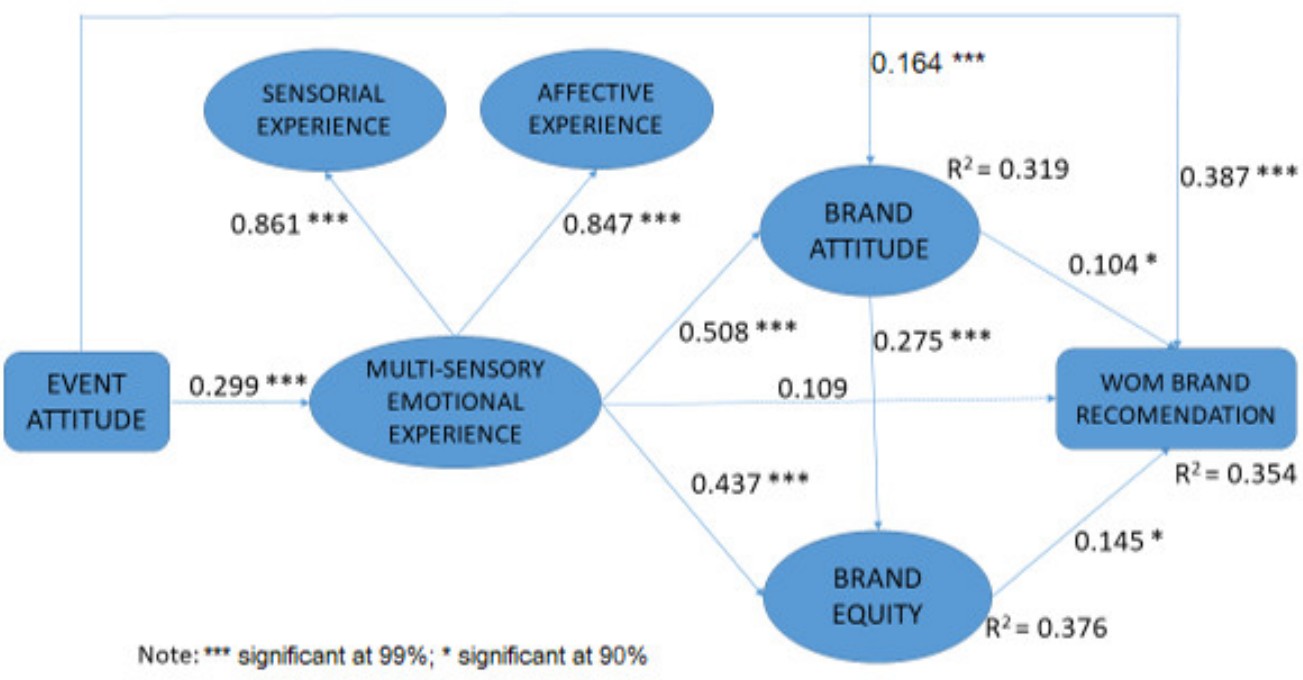

**Figure 2.** SEM Model Results.

## 5. Discussion, Conclusions, Limitations and Future Research

*5.1. Theoretical Implications*

Our analysis sheds light on how experiential events create outstanding experiences that facilitate strong associations with the organizing brand, thereby improving customers' evaluations of said brand. We illustrated this relationship by analyzing the impact of a summer event, organized by an affluent marina, on word-of-mouth (WOM) brand recommendations. To our knowledge, the relationships between event attitude and WOM have not received enough attention in the hospitality literature. Our tested model indicates that more positive evaluations of the event aligned with more positive evaluations and WOM recommendations.

Additionally, the impact of an event must be measured in both the short and the long term, as clients' attitudes and beliefs can manifest in the moment of the experience and then change afterwards [56]. As far as we can tell, there are currently no studies that empirically analyze the effects of experiential events on the organizing brand outside of the immediate moment in which they occur. Therefore, our study's temporal focus is one of its main contributions.

The findings covered a chain of direct and indirect effects that started from the positive evaluation of the event, which was organized using different sensorial stimuli. This event modified visitors' experience, which had an impact on both their post-visit brand attitude and the brand equity. Collectively, the chained effects led to a high probability of recommending the brand.

In terms of direct effects, the event attitude had a positive effect on the multi-sensory emotional brand experience (0.299), which was reflected in the sensorial brand experience (0.861) and affective brand experience (0.847). Both event attitude (0.164) and the multi-sensory emotional brand experience (0.508) had a positive and significant effect on the post-visit attitude toward the brand. The multi-sensory emotional brand experience (0.437) and the favorable attitude toward the brand itself (0.275) had a positive and significant effect on brand equity. Regarding brand recommendation, the post-visit attitude toward the brand (0.387), brand equity (0.145) and brand attitude (0.104) had a positive and significant effect on WOM.

Although the multi-sensory brand experience did not show a significant direct effect on WOM, the standardized indirect effect produced by the combination of previous relationships among constructs was 0.280, as provided by the AMOS output.

*5.2. Managerial Implications*

Our study features some implications for practice. By considering brand experience as a way to measure consumers' response to the brand, marketers may be able to stimulate internal customer responses more holistically, to make events more impactful and thereby facilitate affective and behavioral outcomes [106].

Although the pandemic has affected many service activities, holding events is a key factor in reviving a country's economy [16]. The results obtained in this study, with data taken from a pre-pandemic situation, show how the impact of the event in the company branding strategy might be transferred to a future post-pandemic situation as a way to bolster recovery.

First, we suggest that companies incorporate sensory elements into these events. Brands that can successfully leverage sensory elements can achieve favorable results in terms of outcomes such as attitude, brand equity and WOM recommendations. Second, the results make clear that the event was able to involve visitors in the experience by incorporating strong sensorial components, which produced better recommendations. Although this participation must be voluntary, visitors who decided to participate seemed to experience even greater excitement. In addition, we recommend that such events be carried out repeatedly and not in isolation; more exposure to the brand led customers to assign a greater value to it.

Because our results come from a real environment, rather than a laboratory setting, we are confident that they have practical significance for brand image and brand value discussions. Many practitioners have not yet devised proper ways of measuring an event's effectiveness, since their usual practice is to quantify it through sales or market share. They also usually rely on the number of media appearances, but this indicator does not account for consumers' attitudes. In contrast, this study offers extensive detail about the analyzed event's characteristics in order to illuminate the factors that may have driven its efficacy. Concretely, emotional experience had a strong effect on post-visit brand attitude and equity, and thereby generated better recommendations.

As is well known, many countries have seen their tourism industry impacted by the mobility restrictions provoked by the pandemic. Therefore, companies are trying to organize their activities in order to follow all the safety indications without foregoing the experiential moments that could impart fun and happiness to both residents and tourists. For this reason, our study offers some suggestions for companies looking to develop the type of events described in this study. In this sense, tourism brands might be able to compensate for the strong restrictions on mobility imposed by the coronavirus pandemic with the emphasis on multi-sensory experiential elements. It is not enough to act by converting physical events into virtual ones. If there are complete mobility restrictions and lockdowns, the use of virtual events might prop up the brand and drive the return of visitors once the restrictions are lifted. However, if physical interaction is allowed, brand destination managers and event organizers may incorporate senses such as smell, touch and, if possible, taste, so that the experience provides a remarkable impression. By incorporating these event marketing practices, tourism companies and institutions can more efficiently allocate resources and thereby generate more sustainable economic, social and environmental results.

Regarding the mitigation of health risks posed by COVID-19, brand destinations or specific territories should guarantee a correct implementation of participants' security issues: for instance, by controlling the number of attendees, maintaining security distance among seats, etc. By combining these 'pandemic' impositions with sustainability issues, brands might make great strides in ensuring their visitors' wellness, respecting the natural environment and promoting the local businesses, artists and institutions that constitute the local cultural and social life. For instance, the port has kept on organizing the summer event mentioned, but limited to 800 hundred people and with security distance among seats, personal formation to assure the visitors' wellness, etc. For that, the marina has received the qualification of 'Safe Tourism', from the ICTE—Instituto para la Calidad Turística Española—as a guarantee of a correct implementation of health risk prevention against COVID-19.

Of course, it will take time to establish a new normal. In the meantime, people's hesitation toward crowded festivals, sport activities, etc., will favor the organization of online events that involve different kinds of interactions and sensory elements. Moreover, the virtual broadcast of these events can be a great tool for nurturing a brand's values and supporting the recommendation of new visits and re-visits among tourists and current residents. For this reason, the service sector is becoming more interested in hardware and software developments—such as livestreaming, holographic images, social media and drone surveillance—that allow tourists to enjoy a multi-sensory experience in virtual environments [107]. As more people adopt these disruptive technologies, there are clear opportunities for researchers and practitioners to conceptualize virtual multi-sensory experiences and apply them to event marketing. Enhancing the visitor experience by using innovative technologies could lead to positive outcomes and increase the number of visitors, thereby contributing to the economic aspects of sustainability, which is particularly important for curators [108]. Thus, there is reason to explore the elements of virtual experiences that modulate causation and correlation.

As a final consideration, we would like to stress that domestic tourists and residents can substitute foreigner visitors when international travel is limited, as has been the case

during the COVID-19 pandemic. In this regard, companies can facilitate the recovery of the service sector by promoting events in conjunction with the local government, which can improve certain destinations in a local community in a sustainable manner, and thereby contribute to their long-term resilience.

### 5.3. Limitations and Future Research

Regarding limitations, although our sample size is similar to those of other studies, a larger sample would have nonetheless added reliability and validity to the results. Additionally, the data were gathered in 2017. Although this article explains a mechanism that is valid, reliable and stable along the years, it is advisable to review how it is working in the present day. Moreover, given the profile of those who responded to the survey, the sample could have been skewed toward Spanish visitors relative to other foreign tourists. This asymmetric distribution makes it impossible to analyze the effect of nationality on the achieved results. In addition, before gathering the data, we planned to measure some moderating effects (e.g., residents vs. tourists, degree of exposure, previous experience, sociodemographic factors, among others), but these could not be tested due to the low response rate. Meanwhile, there are possible outcome variables, such as eWOM, that we did not monitor. Given the nature of the analyzed event and the brand's predisposition for encouraging customer engagement on social networks and various platforms, this limitation represents a complementary line of future research. In the future, studies could also strive to analyze participants' behaviors and attitudes across more than one event. Furthermore, we recommend replicating this work with a similar event where the brand is a mere sponsor and not the organizer in order to see if the results shift based on that status.

**Author Contributions:** Conceptualization, M.G.-S., M.J.Y.; methodology, M.G.-S., M.J.Y.; software, M.G.-S.; validation, M.J.Y.; formal analysis, M.G.-S., M.J.Y.; investigation, M.G.-S., M.J.Y.; resources, M.J.Y.; data curation, M.G.-S.; writing—original draft preparation, M.G.-S.; writing—review & editing, M.G.-S.; visualization, M.G.-S., M.J.Y.; supervision, M.J.Y.; project administration, M.J.Y.; funding acquisition, M.J.Y. All authors have read and agreed to the published version of the manuscript.

**Funding:** This research benefited from the Professorship Excellence Program in accordance with the multi-year agreement signed by the Government of Madrid and the Autonomous University of Madrid, UAM (Line #3).

**Institutional Review Board Statement:** The study was conducted in accordance with the Declaration of Helsinki, and followed the protocol issued by the Ethics Committee of Research (Comité de Ética de la Investigación) approved by Universidad Autónoma de Madrid (Spain).

**Informed Consent Statement:** All subjects gave their informed consent for inclusion before they participated in the study.

**Data Availability Statement:** The data presented in this study are not publicly available due to confidentiality and privacy restrictions. They could be made available on reasonable request from the corresponding author with the previous permission of the studied company.

**Acknowledgments:** This research was conducted under the framework of the UAM research group TECHNOCONS 'Consumer Behavior and Technology'. The authors would like to give special thanks to the company and its marketing managers for supporting the data survey. They also appreciate the help from Shawn White, for his great work with the manuscript.

**Conflicts of Interest:** The authors declare no conflict of interest.

## Appendix A. Scale Measurement

| Authors | Constructs | Description |
| --- | --- | --- |
| Avello et al. (2011) | Event Attitude | Unexciting-Exciting |
| | | Boring-Stimulating |
| | | Monotonous-Sensational |
| | | Unappealing-Appealing |
| Reichheld (2006) | WOM recommendation | Probability of recommending this brand to a friend, colleague or relative |
| Zarantonello & Schmitt (2013); Brakus et al. (2009) | Sensorial Experience | I find this brand interesting for discovering new experiences |
| | | This brand makes a strong impression on my visual sense or other senses |
| | | This brand appeals to my senses |
| | Affective Experience | This brand induces my feelings and sentiments |
| | | I have strong feelings for this brand X |
| | | This is a brand which generates positive feelings to me |
| Brunner et al. (2015); Zarantonello & Schmitt (2013) | Brand Attitude | This brand is good |
| | | This brand is pleasant |
| | | Brand X is attractive |
| Zarantonello & Schmitt (2013); Yoo & Donthu (2001) | Brand Equity | Even if any other port in the island has the same features as X, I would prefer to go to this brand |
| | | If there is another port in the island as good as brand X, I prefer to go to this brand |
| | | If another port in the island is not different from brand X in any way, it seems smarter to go to this brand |

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
