# Peer review of "Making Sense from Experience: How a Sustainable Multi-Sensory Event Spurs Word-of-Mouth Recommendation of a Destination Brand"

_sustainability, doi:10.3390/su13115873_

Round 1

Reviewer 1 Report

The paper: "Tell your friends! Promoting a sustainable destination brand through a multi-sensory event" presents an interesting topic.

However, there are some aspects that should be clarified:

  1. The title should be revised. At this moment is not very clear the link between the title and the content.
  2. Please explain all the abbreviations used in the text, and be sure that the same abbreviation is used all along with the manuscript (e. ST, SD).
  3. How representative do you consider your data are at this moment? Since the research was conducted in 2017? Actually, this is my main concern regarding the manuscript.

Good luck!

Reviewer 2 Report

I read and reviewed this paper with great interest. The paper deals with an interesting topic suitable for the Journal. As the authors mention, experiential marketing is of great importance. It is generating a large number of citations within marketing academic research. The manuscript presents a study into how event attitudes and experiences influence brand attitude, equity, and WoM intention towards the brand.

I also find it interesting the fact that the study has been carried out in a European country, with a great tourism industry and in a destination where many foreign residents, especially from Great Britain and France, live and also attracts tourists from different European countries. The study’s focus on the sensorial and emotional dimensions of the experience is a worthwhile consideration. Implications of the research for theory and practice can be clearly identified.

In addition, the paper provides a new multidimensional view of the experiential events. The readability of the paper is to be appreciated. The clarity of expression and the use of the English language seem to be as expected for a scientific publication.

However, there are some minor concerns, which I explain below.

Although it appears that a careful review of the main research element in the study has been carried out. The paper presents a complete theoretical framework, especially given the difficulty of finding recent and relevant work on this topic. But one of my concerns is precisely related to the theoretical funding. The authors developed several hypotheses, but summarize too much the support theories. Developing the theoretical part of the paper, an application of an established theory is a “must” nowadays. Thus, the article will be strengthened by giving more details about the established theories that explains the rest of the background so the research contributions could be more easily seen to the readers.

How was the advertising of the event? How were the sensorial or experiential components? Give more details about them.

The study was carried out before the pandemic provoked by Covid-19 began. A further explanation of Covid impact on the organized event must be added by answering two questions: Has the event been organized in 2020 and if so, how has the company applied the restrictions and security measures?

The use of PLS-SEM instead of CB-SEM is well motivated. The reliability, validity, and structural analysis of the model suggest a robust model. However, only 18.6% were first-time visitors to the port. Would these participants’ attitudes towards the event have different valuations than those that repeated? In other words, is it prior knowledge and experience of the event a determinant of different attitudes?

The results are presented in an adequate and clear manner. In addition, they are in accordance with the theoretical part. It would be advisable to homogenize the values of the results in all the tables and text with the use of either two decimals or three of them. Moreover, include the variation coefficient in Table 1 and explain the dispersion taking into account this statistics.

Round 2

Reviewer 1 Report

The authors improved the manuscript. 

good luck!